**Data Availability Statement:** Our data is under review at Dryad. Our submission has been

# Iterative evaluation of mobile computer-assisted digital chest x-ray screening for TB improves efficiency, yield, and outcomes in Nigeria

Rupert A. Eneogu[1‡], Ellen M. H. Mitchell[2‡*], Chidubem Ogbudebe[3], Danjuma Aboki[4], Victor Anyebe[3], Chimezie B. Dimkpa[3], Daniel Egbule[4], Bassey Nsa[5], Emmy van der Grinten[6], Festus O. Soyinka[7], Hussein Abdur-Razzaq[8], Sani Useni[3], Adebola Lawanson[9], Simeon Onyemaechi[9], Emperor Ubochioma[9], Jerod Scholten[6], Johan Verhoef[6], Peter Nwadike[6], Nkemdilim Chukwueme[10], Debby Nongo[1], Mustapha Gidado[6]

1 United States Agency for International Development (USAID), Abuja, Nigeria, 2 Mycobacterial Diseases and Neglected Tropical Diseases Unit, Department of Public Health, Institute of Tropical Medicine, Antwerp, Belgium, 3 KNCV TB Foundation, Abuja, Nigeria, 4 Nasarawa State TB and Leprosy Control Program, Nasarawa, Nigeria, 5 McMaster University, Ontario, Canada, 6 KNCV TB Foundation, The Hague, Netherlands, 7 Ogun State Ministry of Health, Ogun, Nigeria, 8 Lagos State Ministry of Health, Lagos, Nigeria, 9 National TB and Leprosy Program, Federal Ministry of Health Nigeria, Abuja, Nigeria, 10 New York Medical College, New York, NY, United States of America

‡ RAE, EMHM contributed equally as first authors.
* emitchell@itg.be

## Abstract

Wellness on Wheels (WoW) is a model of mobile systematic tuberculosis (TB) screening of high-risk populations combining digital chest radiography with computer-aided automated detection (CAD) and chronic cough screening to identify presumptive TB clients in communities, health facilities, and prisons in Nigeria. The model evolves to address technical, political, and sustainability challenges. Screening methods were iteratively refined to balance TB yield and feasibility across heterogeneous populations. Performance metrics were compared over time. Screening volumes, risk mix, number needed to screen (NNS), number needed to test (NNT), sample loss, TB treatment initiation and outcomes. Efforts to mitigate losses along the diagnostic cascade were tracked. Persons with high CAD4TB score ($\geq$80), who tested negative on a single spot GeneXpert were followed-up to assess TB status at six months. An experimental calibration method achieved a viable CAD threshold for testing. High risk groups and key stakeholders were engaged. Operations evolved in real time to fix problems. Incremental improvements in mean client volumes (128 to 140/day), target group inclusion (92% to 93%), on-site testing (84% to 86%), TB treatment initiation (87% to 91%), and TB treatment success (71% to 85%) were recorded. Attention to those as highest risk boosted efficiency (the NNT declined from 8.2 ± SD8.2 to 7.6 ± SD7.7). Clinical diagnosis was added after follow-up among those with $\geq$ 80 CAD scores and initially spot -sputum negative found 11 additional TB cases (6.3%) after 121 person-years of follow-up. Iterative adaptation in response to performance metrics foster feasible, acceptable, and efficient TB

assigned a unique digital object identifier (DOI): doi:10.5061/dryad.v9s4mw71p.

**Funding:** The Global Health Bureau, Office of Infectious Disease, US Agency for International Development, financially supported this study through Challenge TB under the terms of Agreement No. AID-OAA-A-14-00029.RAE, EMHM, CO, VA, CBD, BN,EVDG, SU, JS, JV, PN, NC, and MG were all supported by this grant. Employees of USAID are co-authors and contributed to this publication. This publication is made possible by the generous support of the American people through the United States Agency for International Development (USAID). The contents are the responsibility of Challenge TB and do not necessarily reflect the views of USAID or the United States Government.

**Competing interests:** RE and DN are now employed by the funder, but at the time of the study RE worked for KNCV TB Foundation. The funder had a limited role in the review of the manuscript.

case-finding in Nigeria. High CAD scores can identify subclinical TB and those at risk of progression to bacteriologically-confirmed TB disease in the near term.

## Background

The important role of computer-assisted digital chest x-ray screening as a triage tool for identifying people who would benefit from molecular TB diagnostic testing is well documented [1–3]. Although the prospects for a biomarker test in the medium term are good, at present there are no robust triage alternatives to x-ray for most settings [4–6]. While there is growing consensus that digital chest x-ray followed by rapid PCR-based testing is the best option in many settings, there are still uncertainties about how best to operationalize this algorithm in low resource settings [3]. Many high burden settings lack sufficient radiology personnel in the public sector to rapidly interpret high volumes of chest x-rays [3]. Most images in TB screening show no anomalies or very early disease and are thus time-consuming to read and score [7]. Computer-assisted digital chest x-ray scoring technology has potential to rapidly identify those with a high pre-test probability of TB from among a large volume of healthy persons so that a feasible sub-set can be tested.

Historically, the contributions of routine mobile digital chest X-ray TB screening have been challenging to assess because studies rarely fully disaggregate clinical versus bacteriological yields, disclose losses along the diagnostic cascade, or report TB treatment outcomes [8–11]. Publication bias has limited access to results of active case finding pilots with suboptimal risk group targeting, community engagement, yield, or treatment outcomes [12–16]. Policy makers, donors, and community advocates are often hesitant to invest in the infrastructure for mobile digital chest x-ray and GeneXpert MTB/RIF (dCXR/GXP) laboratories due to the limited data on how to maximize and sustain their impact. While guidance has been disseminated, it is rarely realistic to conduct the months of local CAD calibration recommended by experts via costly universal testing with a reference standard [17, 18].

### Evaluation rationale

To address these information gaps and unmet technical needs, we designed rapid and cost-conscious means of setting a sustainable screening algorithm for CAD and retrospectively evaluated routine data. We aimed to assess the viability and field-robustness of optimization strategies to fill this void. Evaluations of routine data are needed because diagnostic accuracy of screening tools are often left censored and promised yields cannot often be achieved in real world settings due to verification bias [10, 19, 20]. This evaluation aimed to make the demands, constraints, costs, and choices facing implementers more explicit.

We report the results of an iterative process to develop and refine TB triage, testing, and treatment algorithm for urban residents of four Nigerian states, in both the southern (Ogun, Lagos) and northern (Kano and Nasarawa) regions, using computer-aided detection (CAD) software program. We describe the steps taken to engage community leaders, define beneficiary groups and participant risk mix, and identify acceptable, feasible, and high transmission settings for high-volume active case finding. We summarize planning processes to ensure yield, flow, confidentiality, and quality of care. The challenges, miscalculations, and debates during all phases are included to highlight operational lessons useful to implementers elsewhere.

## Materials and methods

### Nigerians at-risk of TB

Nigeria is a high TB, TB/HIV, and MDR-TB burden country, with an estimated TB treatment coverage of 24% in 2018 [21]. Home to 9% of the world's missing TB patients, Nigeria has experimented with many models of TB active case-finding (ACF) and results have varied widely [12, 16, 22–25]. Urban Nigeria has an estimated prevalence of bacteriologically-confirmed pulmonary TB between 441–884 cases per 100,000 persons 15 years or older [26]. Prior TB screening efforts suggested considerable heterogeneity, with sustained TB yields obtained only via use of accurate tools and careful attention to participant mix.

Nigeria has tried to increase the effectiveness of screening efforts, moving from a three-week chronic cough threshold to a two-week cough threshold policy in 2014. Symptom driven verbal screening has not increased pulmonary TB (PTB) case notification significantly, even when promoted on a mass scale [23]. While, community referral of chronic coughers is often deemed feasible, due to its potential for community ownership and high coverage, the sensitivity of the classical chronic cough screen has been revised downwards (from 56% to 38%) as consensus on the crucial role of subclinical TB in transmission and endemicity has grown [27–29].

**The Wellness on Wheels (WoW) intervention.** *Infrastructure and staff development.* Two large container trucks housing a lead-lined chest-x-ray suite, reading area, and mobile GeneXpert laboratory were sourced via competitive tendering. Solar panels, shade canopies, and anti-theft adaptations enhanced field-robustness.

CAD4TB by Delft Imaging interprets a digital image in less than a minute and can do so consistently during day-long screening events [30–32]. The software generates a TB likelihood score between 0 and 100, indicating the extent of lung abnormalities. CAD4TB displays a heat-map pinpointing the size and location affected. The scores are used to set a threshold above which persons are invited for TB testing. To determine the threshold, manufacturers and WHO advise that the software should be calibrated in each population and for each version [33].

Each team had a radiology technologist, laboratory technologists, data clerks, truck driver, and a clinical supervisor. Staff exposure to radiation was measured through dosimetry as mandated by the Nuclear Regulatory Agency of Nigeria. Training was conducted on standard operating procedures (SOPs) drawn from procedural manuals of successful digital chest X-ray (CXR) TB screening programs [34]. Experts in social mobilization crafted logos and messaging. The screening intervention was implemented by KNCV TB Foundation (KNCV) under a cooperative agreement, Challenge TB Project, funded by the United States Agency for International Development (USAID) from 2015 to 2019.

**Stakeholder engagement.** Prior to TB screening activity, preparatory visits were made to engage TB stakeholders; sensitize and seek the cooperation of local authorities; assess the feasibility of conducting an effective screening; ensure that all necessary logistics were put in place. The community is mobilized using locally appropriate means such as town announcers, handbills, posters, leaflets, community drama and radio.

Local government TB supervisors, State TB program managers, international and local partners and civil society organizations (CSOs) participated in the inaugural events. Advocacy visits were paid by KNCV ACF teams to key stakeholders in Nasarawa state including the governor, local government chairmen and the traditional chiefs particularly the Emir of Lafia. Community leaders, TB program staff, policy makers, and technical experts helped to select the locations for mobile outreach through a participatory mapping process (Table 1 and S1 File). Explaining the intervention and its objectives and benefits helped meet the expectations of selected communities. The initiative was flagged off at the national level in one of the four

**Table 1. Comparison of TB yield by locations of TB screening and testing events by phase.**

| Phase: | Calibration | Pilot South | | Pilot North | | Pilot Total | | Scale Up South | | Scale Up North | | Scale up Total | | Total screening events | | Mean TB cases per event |
|---|---|---|---|---|---|---|---|---|---|---|---|---|---|---|---|---|---|
| Events: | **8** | **52** | | **46** | | **98** | | **57** | | **60** | | **117** | | **249** | | |
| | **n** | **n** | **%** | **n** | **%** | **n** | **%** | **n** | **%** | **n** | **%** | **n** | **%** | **n** | **%** | $\bar{x}$ (SD) |
| **Poor urban communities** | 6 | 35 | 61 | 31 | 63 | 66 | 62 | 24 | 38 | 30 | 45 | 54 | 42 | 128 | 51 | 1.45(1.86) |
| **Health facilities** | 2 | 4 | 7 | 2 | 4 | 6 | 6 | 0 | 0 | 9 | 13 | 9 | 7 | 17 | 7 | 2.13(1.46) |
| **Silica Factories** | 1 | 1 | 2 | 1 | 2 | 2 | 2 | 4 | 6 | 2 | 3 | 6 | 5 | 9 | 4 | 0.0 (-) |
| **Markets** | 0 | 5 | 9 | 1 | 2 | 6 | 6 | 9 | 14 | 1 | 1 | 10 | 8 | 16 | 7 | 1.78 (1.17) |
| **Prisons** | 0 | 6 | 11 | 11 | 22 | 17 | 16 | 17 | 27 | 19 | 28 | 36 | 28 | 54 | 22 | 2.36 (2.6) |
| **Schools** | 0 | 1 | 2 | 2 | 4 | 3 | 3 | 5 | 8 | 5 | 7 | 10 | 8 | 13 | 5 | 1.6 (1.2) |
| **Motor parks** | 1 | 2 | 4 | 0 | 0 | 2 | 2 | 4 | 6 | 1 | 1 | 5 | 4 | 8 | 3 | 1.67(2.25) |
| **Barracks** | 0 | 0 | 0 | 1 | 2 | 1 | 1 | 0 | 0 | 0 | 0 | 0 | 0 | 1 | 0 | 0.0 (-) |
| **Missing** | 0 | 3 | 5 | 0 | 0 | 3 | 3 | 0 | 0 | 0 | 0 | 0 | 0 | 3 | 1 | 1.60 (-) |
| **Total sites** | **10** | **57** | **100** | **49** | **100** | **106** | **100** | **63** | **100** | **67** | **100** | **130** | **100** | **249** | **100** | **1.72 (2.0)** |

pilot states, Ogun, by the governor supported by the Minister of Health. The flag off brought together different stakeholders including community and religious leaders, community-based organizations, ex-TB patients, different cadre of health workers, security agencies, and political leaders to pledge their support for the intervention.

**Ethical considerations.** This evaluation was conducted at the request of stakeholders using de-identified secondary analysis of routine TB program surveillance data. The retrospective analysis is based upon routine, de-identified surveillance data under the rubric of a data use agreement. The absence of identifying information in the datasets precluded retrospective request for informed consent while simultaneously protecting against risk of harms due to deductive disclosure. The National Health Research Ethics Committee of Nigeria (NHREC) exempted this study from ethical review (NHREC/01/01/2007). Those in the high-risk follow-up cohort were asked for informed consent to re-contact.

**Strategy and prioritization.** Two week-long planning workshops were held involving state, national, community leadership to plan risk group mix, hot spots, testing algorithms, participant flows and crowd management including retention before screening, performance monitoring and targets. Stakeholders voiced diverse assumptions about the effectiveness of computer-assisted CXR interpretation versus cough screening. To reach consensus on a viable and sustainable approach, sensitivity versus specificity of the screening algorithm and screening quality vs quantity trade-offs were debated. Equity concerns were raised when epidemiologists urged a narrow focus on adult men, older adults, alcohol users, urban poor, prisoners and groups with a higher pre-test probability of TB. Debates ensued about the competing demands of reaching high daily screening volumes versus screening those at highest risk, who would be fewer in number. Modelling diverse yield scenarios helped to make strategic choices and manage expectations of donors, TB program managers, and community leaders (See S3 File). The mapping process entailed the identification of groups at higher risk of developing TB disease. These factors included persons sharing similar risk factors for TB such as persons living with HIV and close contacts of pulmonary TB patients or a group of persons living in a specific geographical location associated with high burden of TB (e.g., people living in an urban slum or a prison). For each risk group, an estimate of the population and the proportion that could be reached with the screening service was calculated and the NNS, which is itself a function of the prevalence of TB in the risk group and the screening and diagnostic algorithm. The

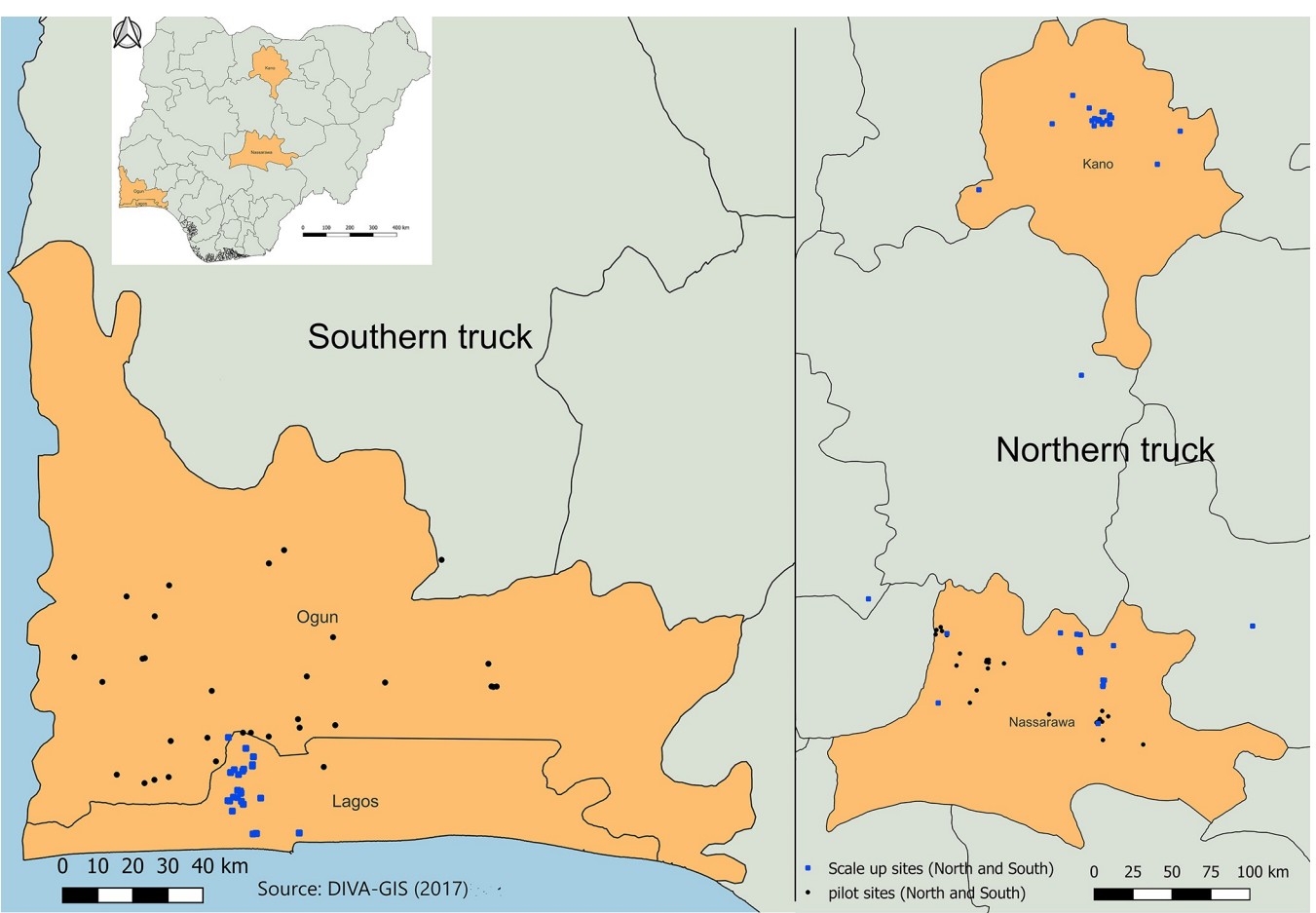

**Fig 1. Geographical distribution of the TB screening activities by phase.** Source of base layers: DIVA-GIS (2017) Free Spatial Data by Country http://www.diva-gis.org/gdata.

community mobilization strategy was geared toward recruitment of men and persons over 30 years of age, because of their elevated vulnerability. Men tend to be less likely to participate in community TB screening, so dedicated efforts are often required to attain a high risk participant mix [26, 35]. Aiming for a high-risk pool with a pulmonary TB prevalence of 1,000/100,000 per population with an estimated 85% sensitivity of the algorithm, we expected an average daily yield of 1.7 persons with bacteriologically-confirmed TB.

The intervention was carried out in three phases across four states (See Fig 1). Findings from each phase informed the design of the subsequent phase. Iterative modification of procedures, strategies, CAD test thresholds and targets occurred after review of prior phase results.

First, a "Calibration phase" was undertaken to identify a feasible CAD4TB score threshold for GeneXpert test eligibility that would ensure a reasonable TB yield given varying micro-epidemic conditions and to ensure value for money. Rigorous calibration would have required TB testing of approximately 30,000 people at low risk over a six-month period, at cartridge cost of 300,000 USD. Instead, a sensitive algorithm comprised of a low testing threshold (CAD4TB score $\geq$ 40) and a classic symptom screen (cough of $\geq$two weeks) were trialed over 8 days (n = 1875). Persons with CAD4TB scores $\geq$ 80 and negative spot TB test results were followed up three to six months later to identify missed TB from GeneXpert MTB/RIF testing on a single spot sample. Emphasis was placed on implementation of a simple algorithm to

facilitate the highest volume screening of highest risk adults while minimizing the participation burden and risk/benefit balance. Pilots were executed in two regions (North, South), to field test the approach. The third phase ("Scale-up") leveraged learning from the calibration and pilots to refine strategy. Persons classified as presumptive for TB followed the national guidelines. Before treatment initiation, a risk factor and clinical interview of bacteriologically confirmed PTB was added to preclude over-diagnosis.

### Data analysis

Acceptability of WOW among stakeholders was pre-defined as willingness to engage by TB program staff, community members, local leaders, and technical stakeholders. Acceptability of CXR screening, cough screening, sputum production, and treatment initiation were all assessed through cascade analysis to measure drop-out or loss of participants at each phase.

Feasibility was defined as a) the ability to screen 1,000 persons per week (200 adults per day over five days), and b) to test those eligible and provide results within 48 hours. To assess the feasibility of different screening and testing algorithms, performance data were collected on total screened daily, proportion screened representing high risk groups, and proportion receiving timely feedback on TB status. Testing frequencies, cartridge costs-per-case, NNS, and NNS inform the choices of score thresholds to balance algorithm sensitivity, feasibility of lab throughput. Sub-group comparisons were conducted to identify recruitment and testing problems and yield in higher risk groups (e.g. men, people over 30 years, incarcerated people).

Fidelity to the intervention design was designed as adherence to the SOPS and field manuals after training. Fidelity was measured via monitoring visits and interim data queries.

The yield of the intervention was defined as the prevalence of bacteriologically confirmed PTB among the population screened, as well as the prevalence of rifampin-resistant TB among the population screened. The efficiency of the intervention was defined as the NNS and NNT to diagnose one bacteriologically-confirmed PTB case [36].

Participants in the calibration and southern pilot with CAD4TB score ≥80 and a negative GeneXpert MTB/RIF result were followed up two to six months after screening to assess the volume and pace of TB progression.

## Results

The outputs and outcomes of TB screening are described in three phases to highlight the impact of the evolving strategy and procedures. The distribution of screening sites is described in Table 1 and performance metrics by phase are summarized in Table 2 and S3 File.

### Calibration phase (8 events)

Urban slums and health facilities in three local government areas of Ogun state were selected to test the suitability and viability of the algorithm, a test threshold of ≥60, and effectiveness of training on SOPs. In the Calibration phase recruitment targets were met, with an average of 203 (101%) persons ≥ 15 years screened per day (Table 2). As planned, 86% of those screened were over thirty years of age. However, the calibration cohort mix skewed female (62%). Acceptability of chest x-ray screening was high (99.5%). As intended, during calibration, a large proportion of participants (41.0%) were flagged for testing for having a CAD4TB score of ≥40. Cough of two weeks duration or more flagged 14.7%, but 97.6% of chronic coughers also had qualifying CAD4TB scores. The high sensitivity of the calibration screen yielded 43.4% eligibility for TB testing. The 769 presumptive clients, an average of 96 per day, exceeded the capacity of the two 4-module GeneXpert machines (32 tests/day). Local laboratory and TB treatment facilities were engaged to help handle the overflow of samples and presumptive

**Table 2. Comparison of key performance metrics by phase.**

| Phase | Pilot | | Scale Up | | p-value |
|---|---|---|---|---|---|
| CAD4TB Test threshold | ≥ 60 or <0 | | ≥ 60 S and ≥ 57 N | | |
| sample | n = 11,280 | | n = 16,637 | | |
| Total days of screening | 98 | | 118 | | |
| **Acceptability in Population** | n | % | n | % | |
| Average daily screen (SD) | 128 | **SD** | 140 | **SD** | |
| Age over 30 years | 8771 | **77.8** | 11380 | **68.4** | |
| Male gender | 6552 | **58.1** | 10971 | **65.9** | |
| higher demographic risk for TB | 10316 | **91.5** | 15465 | **92.9** | 0.00003 |
| lower demographic risk for TB | 964 | **9%** | 1172 | **7%** | |
| Screened for chronic cough | 11279 | **100** | 16636 | **100** | |
| Screened with CXR | 11237 | **99.6** | 16615 | **99.9** | |
| Interpretable image | 11119 | **98.9** | 16540 | **99.1** | |
| Presumptive by chronic cough only | 295 | **2.6** | 439 | **2.6** | |
| Presumptive by CAD4TB threshold | 1542 | **13.7** | 1091 | **6.6** | |
| Presumptive: eligible | 1917 | **17.2%** | 1527 | **10.6** | |
| **Programmatic Fidelity** | | | | | |
| Sample not produced, low quality, not tested | 301 | **15.7** | 213 | **13.9** | 0.15 |
| Tested for TB with GXP on-site among eligible | 1616 | **84.3** | 1314 | **86.1** | |
| Bacteriologically confirmed TB | 196 | **12%** | 169 | **13%** | |
| negative spot GeneXpert MTB/RIF | 1420 | **88%** | 1145 | **87%** | |
| RIF resistant TB | 4 | **2%** | 7 | **4%** | |
| Clinically diagnosed TB | 0 | **0.0** | 3 | **1.7** | |
| HIV tested | 151 | **77%** | 113 | **65%** | |
| No HIV test | 45 | **23%** | 59 | **34%** | 0.02 |
| TB Treatment initiation | 170 | **87%** | 158 | **91%** | |
| No TB treatment initiated | 26 | **13%** | 14 | **8%** | 0.15 |
| Treatment success | 120 | **71%** | 134 | **85%** | |
| Unfavorable treatment outcome | 50 | **29%** | 38 | **24%** | 0.06 |

clients. Over 35% of samples had to be tested off-site or outside the target window of 48 hours. Moreover, 15% of presumptive clients could not spontaneously expectorate a spot sputum sample to test. Electrical interruptions gave high error rates (6%) and obliged re-testing of samples. Fidelity to the data management SOPs was low and data linkage was manual. Two persons with bacteriologically confirmed PTB were identified among 1,875 adults screened, yielding an NNS of 958, and a NNT of 385. GeneXpert cartridge costs per confirmed TB case were US$3,842 (385*$9.98) which was deemed unsustainably high. The TB patients identified had high CAD4TB scores (>95). More than a quarter (27%) of the general population had CAD4TB scores between 40 and 59 but testing in this group yielded no cases. So, stakeholders urged an increase in specificity to a level that matched the on-site TB testing capacity (max of 32 tests/day) of the trucks. Calibration was viewed as a poor use of limited resources and a threat to project reputation by some stakeholders who instead urged a more traditional focus on detection of TB in symptomatic individuals.

## Pilot phase (98 events)

Piloting of a revised threshold was conducted over 98 screening days (Table 2). Participation in screening averaged 64.0% of the 200 per day target. A greater proportion of adult men

(58.1%) participated in better alignment with Nigeria's disproportionately male TB epidemic [21]. Fig 2 describes the testing algorithm in the Pilot Phase. The CAD4TB threshold score for TB testing was raised from ≥40 to ≥60 resulting in a viable proportion of presumptive TB clients (41.0% to 13.7%). Those with cough ≥ 2 weeks but with a CAD4TB score lower than the threshold remained eligible for TB testing and contributed 2.6% to testing volumes. Construction of a quasi 'receiver operating curve' combining the calibration and pilot data suggested by lowering the testing threshold to ≥56, we could achieve gains of ≈1–3% in sensitivity at a cost of ≈2–4% specificity and 3% testing. As the proportion selected for TB testing (11% or 22 tests/day) in the North was well below that of the laboratory capacity (16% or 32 tests/day) a more sensitive test threshold was deemed viable in the North.

As the volume of people to be tested declined, cascade retention improved; missing or delayed results declined from 58% to 15% (Table 2) Bacteriologically confirmed TB yield increased eight-fold from the calibration from 0.2% (2/1875) to 1.7% (196/ 11,279) of participants.

Fidelity to data management and laboratory SOPs improved and error rates declined from 6% during calibration to 0.5% in the pilot. Personal identifiers were only collected if a person screened positive, to focus on completeness of sample collection, yield, and turnaround time. In all, 86.7% of those diagnosed started treatment,77% were tested for HIV, and 70.6% had a documented successful outcome.

**Follow-up of high CAD score with negative GeneXpert.** There were 225 people who scored ≥ 80 on CAD4TB yet tested negative on a single spot GeneXpert and were eligible for follow-up at six months to re-assess TB status. A total of 173 (77%) were reached by phone or home visit. They were screened for TB symptoms and invited to re-test for TB an average of 242 days (range 52–358) after the original screen. Eighty-nine (51%) agreed to re-test, could produce a sample, and four (4.4%) tested positive for TB. Fifty-one (23%) could not produce a sample. Four (2.3%) refused. Three (1.7%) had died, including one from TB (0.6%). Six (3.4%) had been diagnosed with TB in the interim and begun treatment. In total 11 persons with TB (6.3%) were identified during 121 person-years of follow-up. The mean time to TB detection was 296 days (95%CI 286–306). Six of 11 people (55%) detected with TB during follow-up had CAD4TB scores of 99 or 100 during initial screening. The elevated risk of disease found via follow-up led to a new policy of on-site clinical diagnostic review of all participants with CAD4TB scores ≥ 80 regardless of initial molecular test result.

## Scale-up phase (118 events)

The scale up period lasted 117 days. The average daily screening volume increased from 64.0% to 70.5% of the 200/day/team target (Table 2). Demographic outreach remained appropriate overall, with 92.9% considered elevated risk (65.9% male, 68.4% over 30 years of age). Acceptability and interpretability of chest x-ray remained high (>99%). The lowered testing threshold in the North (CAD4TB score ≥56) increased the proportion of presumptive clients by 1.4% (n = 240). However, the overall proportion of those screening positive declined precipitously from 16.3% to 10.6% (1,527/16,636) although there was no change in the testing threshold.

The yield of TB declined during scale up from 1.7% to 1.0% despite a consistent site mix of roughly a quarter (22–28%) of days spent at prisons, half of days (62–42%) spent in poor urban centers, and the remaining spent in hotspots such as health centers (6%-7%), markets (6–8%), and motor parks (2%-4%) (See Table 1). The NNS to diagnose one case almost doubled from 58 to 98 during scale up, while NNT declined slightly from 7.8 during the pilot to 8.2, suggesting improved efficiency (Table 3).

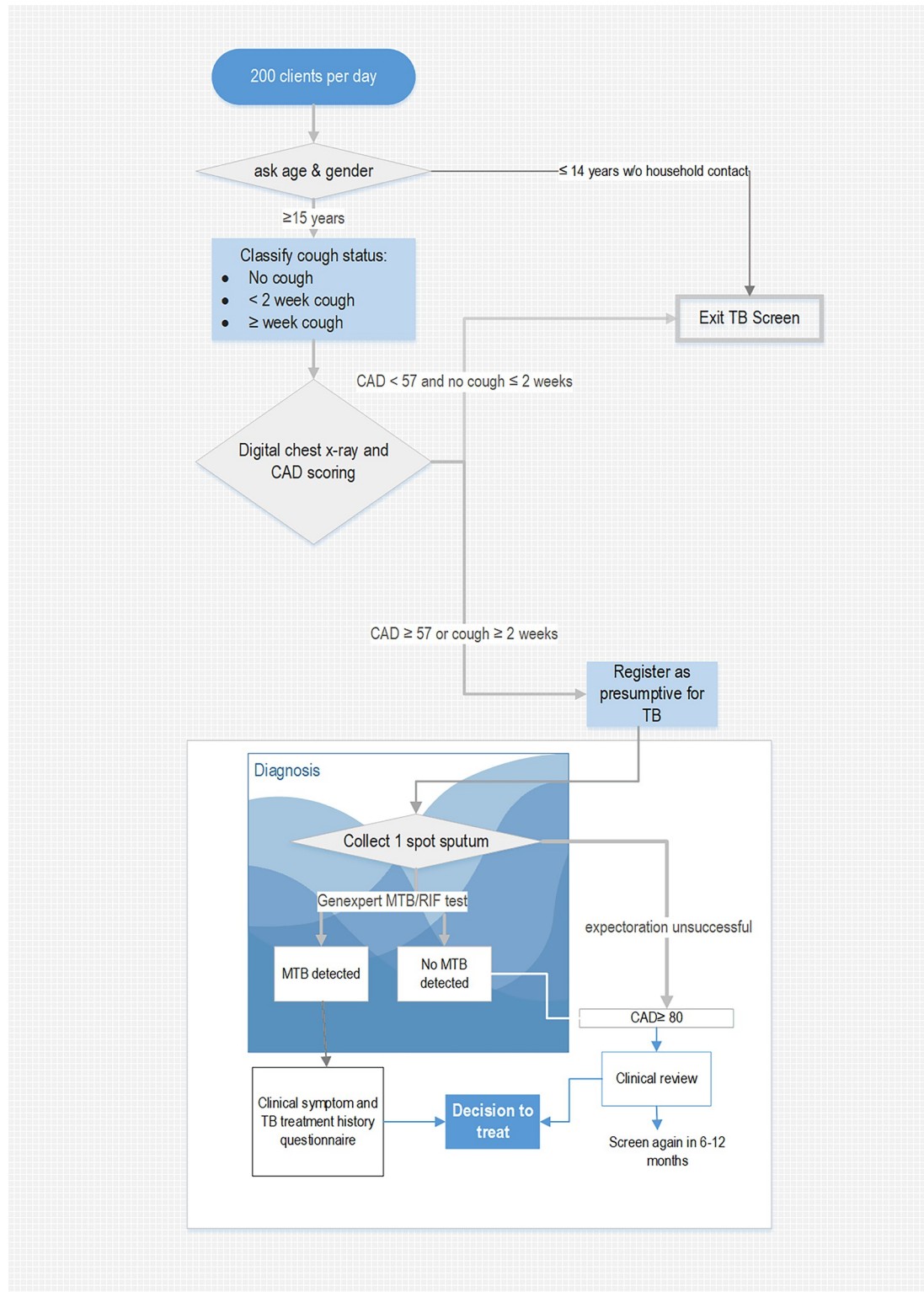

**Fig 2. Client flow and testing algorithm in pilot and scale Up phases.**

**Table 3. Efficiency of active case finding by phase.**

|  | Calibration | Pilot | Scale-Up |
|---|---|---|---|
| Prevalence of bacteriologically-confirmed TB per 100,000 screened | 600 | 1700 | 1022 |
| Average daily yield of TB diagnoses | 0.25 | 2 | 1.5 |
| No. needed to screen (NNS) to detect 1 bacteriologically confirmed patient | 958 | 58 | 98 |
| No. needed to test (NNT) detect 1 bacteriologically confirmed patient | 385 | 8.2 | 7.6 |
| Cartridge costs per TB case (US$) | $3,842 | $82 | $76 |

In the scale-up period, the southern truck moved from Ogun state to Lagos state. The trucks were too heavy and wide to navigate the narrow paths of informal settlements built on land reclaimed from the sea. Lack of access to the poorest neighborhoods may explain the increase in NNS from 42 to 86 in the South. Another possibility is that the proportion of days screening near schools and factories increased slightly, and these locations had lower yield. HIV co-infection among TB patients declined from 3.5% to 2.5%.

There were 171 persons over 15 years of age diagnosed with PTB in the scale-up phase. The CAD4TB threshold ≥60 identified 92.9% (159/171) of those with bacteriologically confirmed TB patients, 44 people (26.0%) screened positive for both chronic cough and chest x-ray abnormalities, and 12 (7.1%) screened positive via chronic cough alone. The lowered test threshold of ≥56 CAD score in the North, detected two (2.2%) additional asymptomatic patients (See S3 File). Three people were diagnosed clinically via chest x-ray interpretation. Approximately 66% (115/174) of PTB identified by GeneXpert was not accompanied by traditional chronic cough. In-depth clinical interviews prior to treatment initiation identified 10 persons treated for TB within the previous two years, requiring physician confirmation of active disease.

Leveraging lessons from the earlier phases, data and sample management practices were refined and although the prevalence in the population declined, TB testing became more efficient, with NNT declining from 8.2 to 7.6 even as the underlying prevalence decreased (Table 3).

As summarized in Table 4, new monitoring and evaluation tools, variable standards, and data linkage methods were introduced to facilitate data-driven decision making in real-time. Barcodes for unique identifiers and auto-fill were introduced to expedite the process and minimize sample loss. Staff were retrained on revised SOPs and databases revised to expedite sample and client tracking. Fidelity to intervention design, data management, and SOPs increased, reducing missing samples from 15.7% to 13.9% during scale-up. Linkage to treatment increased from 86.7% to 90.8% and successful treatment outcomes increased from 70.6% to 83.2%. However, HIV test coverage decreased to 64.9% from 77.0% due to intermittent commodity stock outs. Approaches to address the challenges are summarized by phase in Table 4.

## Discussion

Over the course of three phases, the WOW project tested models, iteratively identified and overcame challenges, refined strategies in collaboration with stakeholders, and tracked outcomes. Monitoring the key metrics to limit losses along the screening cascade helped gain efficiencies with NNT, cartridge utilization, treatment initiation, and outcomes over time. Target groups and CAD4TB score testing thresholds should be regularly adjusted to maximize case-finding. We found that challenges were often team-specific; with operational issues such as crowd control, unprocessed sputum sample backlog, difficult terrain, and highly variable proportions of presumptive TB over time. Varying screening criteria and thresholds permitted the

**Table 4. Challenges and solutions by intervention phase.**

| Phase | Challenge | Remedy implemented |
|---|---|---|
| **Calibration** | Stakeholders dissatisfied with high testing proportion, poor sample management, and perceived low bacteriologically confirmed TB yield | Practice good communication about goals and limit duration of threshold calibration, expectation management is crucial for ACF |
| | Low testing threshold (≥40 CAD4TB score), identifies approximately 40% of population as presumptive TB case. | Construct censored ROC curves and cost estimation to identify viable threshold that maximizes yield without exceeding test capacity |
| | Low testing threshold (≥40 CAD4TB score) identifies many asymptomatic individuals who could not produce a testable sputum sample | Raise the CAD threshold for testing to ≥60, while maintaining the specific symptom (chronic cough of two week duration) to avoid under-detection. Use an inexpensive, field-safe induction method. Provide sputum coaching using evidence-based methods |
| | Linking participants' symptom screen, CAD4TB, and testing data is time consuming and error prone due to lack of Unique ID | Use barcoded unique ID to match symptoms, socio-demographics, CAD scores, presumptive registry, GXP results, and treatment outcomes |
| | Vigorous social mobilization leads to crowd control challenges, delays, skewed participant-mix | Truck driver performs crowd control function. Participants receive a time window |
| | Laboratory sample management, testing suboptimal, GeneXpert machine has 6% error rate | Retrain staff on procedures for GeneXpert MTB/RIF machine use. Stabilize electricity |
| **Pilot** | Participant follow-up indicates single spot specimen for GeneXpert MTB/RIF misses some early TB | Add clinical diagnosis of high CAD4TB scores. Refer those with CAD4TB >80 for follow-up. Follow-up persons with a positive CAD (score >60) and a negative GeneXpert MTB/RIF |
| | Adolescents (10–17 years) and persons with fibrotic lesions often scored negative CAD4TB values, but positive on GeneXpert MTB/RIF | Collect sputum for testing among those with negative CAD4TB scores. Add clinical diagnosis of negative CAD4TB scores |
| **Scale-up** | TB hotspots (prisons 3.6 cases/day, motor parks 2.5/day) are limited in number and size of target population. Poor communities have more potential participants but lower TB prevalence (1.8 cases/day) | Use micro-targeting to influence participant mix: Combine TB screening with services that are more highly valued by highest risk groups (e.g., vision, hearing screenings to increase engagement of older people). Actively monitor yield with weekly dashboards and use to make decisions on when to relocate |
| | High volume screening is logistically intensive and difficult to sustain. Staff burnout, fatigue, and stress are risks | Rotate staff, provide security, limit shift length. Conduct time-motion studies to maximize efficiency, reduce wait times, and limit crowding |
| | Initiating TB treatment in asymptomatic people based on chest x-ray and GeneXpert MTB/RIF findings can lead to over-treatment | Avoid over-diagnosis of people with previous TB by instituting a clinical exam and history before making a definitive TB diagnosis [28] |
| | Matching screening and testing data are still not seamless even with barcodes | Develop application programming interface (API)s to integrate data from CAD and GeneXpert machines. |

comparison of the performance of individual screening methods (chronic cough vs CAD4TB vs combined) and thresholds (≥ 40 vs. ≥ 60). CAD4TB score thresholds for TB testing with Xpert should be set high enough so that sample throughput can be managed on-site to preclude diagnostic delay. Calibration indicated that a test threshold of ≥60 (or negative) and/or cough of two-week duration was a sustainable algorithm in the South, whereas ≥56 (or negative) and/or cough of two-week duration was a viable algorithm in the North.

We show that high-yield TB case finding campaigns using expensive equipment requires attention to the socio-demographic and risk-factor mix in the intended screening pool. Establishing participant mix goals reflective of the gender and age distribution of the national epidemic proved valuable. Equally crucial was detailed community mapping and early engagement of stakeholders in prisons, health facilities, and workplaces. Close collaboration with community organizations and the TB program helped minimize the treatment initiation challenges reported in some TB ACF interventions driven by non-governmental organizations [35]. A rich site mix of prisons, out-patient departments, dense urban low-income communities, and male-dominated workplaces favored a high risk pool, but did not guarantee it. The high specificity of the screening algorithm was strategic and reduced excess testing in low burden sites. Further analyses are needed to explain spatial variation in yield.

Despite a conscious community engagement strategy, daily screening targets of 200 clients were not routinely achieved. Our evaluation highlights the challenge of sustaining ambitious daily screening volumes year-round in risk groups of finite size and spatial dispersion.

As is common, this mobile digital chest x-ray and GeneXpert screening effort required active management of diverse stakeholder expectations [13, 14]. Pressures to demonstrate return on investment in the short term meant little appetite for full calibration due to cost and efficiency concerns. Moreover, the high fixed and running costs of digital chest x-ray and mobile GeneXpert labs made detection of two or three people with TB per day seem disappointing to some stakeholders, even though this is among the highest yielding community active case finding efforts ever conducted in Nigeria [23, 24, 37, 38]. Digital chest x-ray with CAD appears to triage better in Nigeria than in contexts with aging populations, higher HIV, smoking, and diabetes prevalence, detecting 87% of TB diagnosed [30, 32, 39, 40].

However, losses due to inability to expectorate are common in active case-finding with digital chest x-ray, but under-reported [35, 41, 42]. Inability to expectorate was historically equated with lower TB disease risk, but evidence no longer supports this posture [10, 43, 44]. A limitation of this evaluation and others is the sparse insight into reasons for missing sputum samples, an under-researched challenge [10, 45, 46]. Innovative sampling solutions for asymptomatic clients with high likelihood of disease on CAD will be needed for the full impact of computer-assisted digital chest x-ray screening and sensitive diagnostics to be realized [47–50]. Over-reliance on a single spot sample missed early TB due to limitations of sample quality, quantity, and test performance [51]. We found annual risk of progression to TB disease ranged from 2–6% among those with chest x-ray abnormalities and negative spot GXP. CAD scores should be used to pinpoint people who need human clinical review and early intervention to treat clinical disease and avert imminent TB [52]. Our progression findings are similar to two Asian studies: in 1979 trialists in Hong Kong documented active disease in 93 (53%) of 176 bacteriologically negative, CXR-positive participants in the small control arm of a chemotherapy study, 75 (43%) during the first year of follow up [53]. In 2008, a population-wide study in Cambodia showed progression to TB in a similar group was 8.5% (95% CI 6.3–11.2) annually [54].

These findings suggest that the return on investment from CAD may be doubled by following up clients with high CAD scores and/or provision of an early intervention regimen or vaccine. Trials leveraging the predictive value of CAD and viability of shorter TB drug regimens for incipient subclinical disease are needed.

This intervention highlights the importance of ensuring that the basics of TB screening, sample collection, testing, and TB treatment linkage are operating well before adding complementary services. A stepwise approach involving meeting screening volume targets, ensuring sample collection, quality-assured TB testing, and initiating TB treatment was sufficiently challenging at the onset. Household contact investigation, risk-group specific algorithms, screening for co-morbidities, clinical diagnoses of extrapulmonary TB, and follow-up or interventions for those at highest risk of progression can be progressively incorporated, once the quality and timeliness of core TB screening functions are established [3].

Innovations in CAD software now leverage accumulated information on digital radiographs to improve the prediction of the software in a process called deep learning [55, 56]. We show that applying a similar 'deep learning' logic to the entire process of TB screening can enhance outcomes.

## Conclusion

Implementation research in two settings with an evolving strategy shows the key success factors for mobile digital chest X-ray and GeneXpert (dCXR/GXP) case-finding in urban Nigeria.

Striking a sustainable balance between sensitivity and feasibility in mobile dCXR/GXP screening requires data-driven adaptation to respond to both the heterogeneity of TB micro-epidemics and the known socio-demographic correlates of risk. To maximize feasibility, acceptability, quality, and yield of ACF over time, continuous review and re-calibration of outreach strategies, algorithm, and procedures were essential. To build effective screening and testing thresholds and algorithms, early pilot efforts require experimentation and regular course correction.

## Supporting information

**S1 File. Mobile dCXR/GXP planning workshop report.**
(PDF)

**S2 File. Wellness on Wheels (WOW) active case finding monitoring and evaluation plan.**
(PDF)

**S3 File. Regional and sensitivity analyses.**
(PDF)

## Acknowledgments

We acknowledge the important contributions of Nigerian community leaders and advocates including Dr. Baba Gana Adams, Mrs Ibiyemi Fakande, and Cecilia Kafran in planning and accountability monitoring. The contributions of World Health organization staff including Omoniyi Fadare and Enang Oyama are similarly appreciated. We are grateful to Amy Piatek and Mukadi Ya Diul and others at the United States Agency for International Development (USAID) for technical support. We appreciate Rian Snijders' assistance with map making. We acknowledge the contributions and comradery of our departed colleague Bashir P. Zakariyya who worked tirelessly to bring the WOW project to fruition.

## Author Contributions

**Conceptualization:** Rupert A. Eneogu, Ellen M. H. Mitchell, Chidubem Ogbudebe, Danjuma Aboki, Festus O. Soyinka, Hussein Abdur-Razzaq, Jerod Scholten, Peter Nwadike, Nkemdilim Chukwueme, Debby Nongo.

**Data curation:** Rupert A. Eneogu, Ellen M. H. Mitchell, Chidubem Ogbudebe, Danjuma Aboki, Victor Anyebe, Chimezie B. Dimkpa, Festus O. Soyinka, Hussein Abdur-Razzaq, Peter Nwadike, Nkemdilim Chukwueme.

**Formal analysis:** Rupert A. Eneogu, Ellen M. H. Mitchell, Chidubem Ogbudebe, Hussein Abdur-Razzaq, Nkemdilim Chukwueme.

**Funding acquisition:** Rupert A. Eneogu, Festus O. Soyinka, Hussein Abdur-Razzaq, Emperor Ubochioma, Jerod Scholten, Johan Verhoef, Nkemdilim Chukwueme, Debby Nongo, Mustapha Gidado.

**Investigation:** Rupert A. Eneogu, Chidubem Ogbudebe, Festus O. Soyinka, Jerod Scholten, Peter Nwadike, Nkemdilim Chukwueme.

**Methodology:** Rupert A. Eneogu, Ellen M. H. Mitchell, Danjuma Aboki, Victor Anyebe, Chimezie B. Dimkpa, Festus O. Soyinka, Hussein Abdur-Razzaq, Sani Useni, Jerod Scholten, Peter Nwadike, Nkemdilim Chukwueme, Mustapha Gidado.

**Project administration:** Rupert A. Eneogu, Chidubem Ogbudebe, Danjuma Aboki, Victor Anyebe, Daniel Egbule, Bassey Nsa, Emmy van der Grinten, Festus O. Soyinka, Hussein

Abdur-Razzaq, Sani Useni, Adebola Lawanson, Simeon Onyemaechi, Emperor Ubochioma, Jerod Scholten, Johan Verhoef, Peter Nwadike, Nkemdilim Chukwueme, Mustapha Gidado.

**Resources:** Danjuma Aboki, Victor Anyebe, Bassey Nsa, Emmy van der Grinten, Sani Useni, Adebola Lawanson, Peter Nwadike, Mustapha Gidado.

**Software:** Chidubem Ogbudebe, Victor Anyebe, Chimezie B. Dimkpa, Mustapha Gidado.

**Supervision:** Chidubem Ogbudebe, Danjuma Aboki, Victor Anyebe, Chimezie B. Dimkpa, Bassey Nsa, Emmy van der Grinten, Hussein Abdur-Razzaq, Sani Useni, Adebola Lawanson, Jerod Scholten, Peter Nwadike, Mustapha Gidado.

**Validation:** Ellen M. H. Mitchell, Chidubem Ogbudebe, Emmy van der Grinten, Adebola Lawanson, Simeon Onyemaechi, Peter Nwadike.

**Visualization:** Ellen M. H. Mitchell, Chidubem Ogbudebe.

**Writing – original draft:** Rupert A. Eneogu, Ellen M. H. Mitchell.

**Writing – review & editing:** Rupert A. Eneogu, Ellen M. H. Mitchell, Chidubem Ogbudebe, Danjuma Aboki, Victor Anyebe, Chimezie B. Dimkpa, Daniel Egbule, Bassey Nsa, Emmy van der Grinten, Festus O. Soyinka, Hussein Abdur-Razzaq, Sani Useni, Adebola Lawanson, Simeon Onyemaechi, Emperor Ubochioma, Jerod Scholten, Nkemdilim Chukwueme, Debby Nongo, Mustapha Gidado.

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
