## [Decision Letter · Decision Letter 0]

10 Apr 2023

PGPH-D-23-00201

Iterative evaluation of mobile computer-assisted digital chest x-ray screening for TB improves efficiency, yield, and outcomes in Nigeria

Dear Dr. Mitchell,

Thank you for submitting your manuscript to PLOS Global Public Health. After careful consideration, we feel that it has merit but needs some minor revisions to fully meet PLOS Global Public Health’s publication criteria . Therefore, we invite you to submit a revised version of the manuscript that addresses the points raised during the review process.

Please review the comments by the editor, reviewer 1 and reviewer 2 to be fully able to revise your manuscript

We look forward to receiving your revised manuscript.

Kind regards,

Shifa S. Habib

Academic Editor

Journal Requirements:

1. You indicated that ethical approval was not necessary for your study. We understand that the framework for ethical oversight requirements for studies of this type may differ depending on the setting and we would appreciate some further clarification regarding your research. Could you please provide further details on why your study is exempt from the need for approval and confirmation from your institutional review board or research ethics committee (e.g., in the form of a letter or email correspondence) that ethics review was not necessary for this study? Please include a copy of the correspondence as an ""Other"" file.

2. Please send a completed 'Competing Interests' statement, including any COIs declared by your co-authors. If you have no competing interests to declare, please state "The authors have declared that no competing interests exist". Otherwise please declare all competing interests beginning with the statement "I have read the journal's policy and the authors of this manuscript have the following competing interests:"

3. Please amend your detailed Financial Disclosure statement. This is published with the article. It must therefore be completed in full sentences and contain the exact wording you wish to be published.

4. Please note that your Data Availability Statement is currently missing  the DOI/accession number of each dataset OR a direct link to access each database. If your manuscript is accepted for publication, you will be asked to provide these details on a very short timeline. We therefore suggest that you provide this information now, though we will not hold up the peer review process if you are unable.

5. Please provide separate figure files in .tif or .eps format and remove the embedded figures from the manuscript file.

6. Some material included in your submission may be copyrighted. According to PLOS’s copyright policy, authors who use figures or other material (e.g., graphics, clipart, maps) from another author or copyright holder must demonstrate or obtain permission to publish this material under the Creative Commons Attribution 4.0 International (CC BY 4.0) License used by PLOS journals. Please closely review the details of PLOS’s copyright requirements here: PLOS Licenses and Copyright. If you need to request permissions from a copyright holder, you may use PLOS's Copyright Content Permission form.

Potential Copyright Issues:

Figure 2: please (a) provide a direct link to the base layer of the map (i.e., the country or region border shape) and ensure this is also included in the figure legend; and (b) provide a link to the terms of use / license information for the base layer image or shapefile. We cannot publish proprietary or copyrighted maps (e.g. Google Maps, Mapquest) and the terms of use for your map base layer must be compatible with our CC-BY 4.0 license. 

Editor Comments:

In the introduction, it will be useful to provide a few lines (2-3) on the implementation experience in other high TB burden countries, including its impact and feasibilityStrengthen the rationale by explicitly stating the evidence gaps that this study will be fulfillingDescribe the implementation settings eg health facilities, schools, factories and the rationale for their selection (if any)Discuss concordance/discordance of specific findings with other studies in the discussion section

Reviewers' comments:

Reviewer's Responses to Questions

**Comments to the Author**

1. Does this manuscript meet PLOS Global Public Health’s publication criteria? Is the manuscript technically sound, and do the data support the conclusions? The manuscript must describe methodologically and ethically rigorous research with conclusions that are appropriately drawn based on the data presented.

Reviewer #1: Yes

Reviewer #2: Yes

2. Has the statistical analysis been performed appropriately and rigorously?

Reviewer #1: I don't know

Reviewer #2: I don't know

3. Have the authors made all data underlying the findings in their manuscript fully available (please refer to the Data Availability Statement at the start of the manuscript PDF file)?

Reviewer #1: Yes

Reviewer #2: No

4. Is the manuscript presented in an intelligible fashion and written in standard English?

Reviewer #1: Yes

Reviewer #2: Yes

5. Review Comments to the Author

Reviewer #1: A clear and reasoned presentation of an important approach to implementation and scale up. This is a useful description of an interative process to utilize new technology effectively. I appreciated the attention to community involvement and engagement.

Reviewer #2: Some data is not fully available, however the authors have stated it can be made available with the consent of the NTP, Nigeria.

The language in many instances is confusing and not well explained, I would recommend editing accordingly.

6. PLOS authors have the option to publish the peer review history of their article (what does this mean?). If published, this will include your full peer review and any attached files.

**Do you want your identity to be public for this peer review?** For information about this choice, including consent withdrawal, please see our Privacy Policy.

Reviewer #1: No

Reviewer #2: No

---

## [Editor Report · Decision Letter 1]

29 Nov 2023

Iterative evaluation of mobile computer-assisted digital chest x-ray screening for TB improves efficiency, yield, and outcomes in Nigeria

PGPH-D-23-00201R1

Dear Mitchell,

We are pleased to inform you that your manuscript 'Iterative evaluation of mobile computer-assisted digital chest x-ray screening for TB improves efficiency, yield, and outcomes in Nigeria' has been provisionally accepted for publication in PLOS Global Public Health.

Best regards,

Shifa S. Habib

Academic Editor
